

# Exploring the classification of cancer cell lines from multiple omic views

Xiaoxi Yang[1], Yuqi Wen[1], Xinyu Song[2], Song He[1] and Xiaochen Bo[1]

[1] Department of Biotechnology, Beijing Institute of Radiation Medicine, Beijing, China
[2] Key Laboratory of Biomedical Engineering and Translational Medicine, Ministry of Industry and Information Technology, Chinese PLA General Hospital, Beijing, China

## ABSTRACT

**Background**. Cancer classification is of great importance to understanding its pathogenesis, making diagnosis and developing treatment. The accumulation of extensive omics data of abundant cancer cell line provide basis for large scale classification of cancer with low cost. However, the reliability of cell lines as in vitro models of cancer has been controversial.

**Methods**. In this study, we explore the classification on pan-cancer cell line with single and integrated multiple omics data from the Cancer Cell Line Encyclopedia (CCLE) database. The representative omics data of cancer, mRNA data, miRNA data, copy number variation data, DNA methylation data and reverse-phase protein array data were taken into the analysis. TumorMap web tool was used to illustrate the landscape of molecular classification. The molecular classification of patient samples was compared with cancer cell lines.

**Results**. Eighteen molecular clusters were identified using integrated multiple omics clustering. Three pan-cancer clusters were found in integrated multiple omics clustering. By comparing with single omics clustering, we found that integrated clustering could capture both shared and complementary information from each omics data. Omics contribution analysis for clustering indicated that, although all the five omics data were of value, mRNA and proteomics data were particular important. While the classifications were generally consistent, samples from cancer patients were more diverse than cancer cell lines.

**Conclusions**. The clustering analysis based on integrated omics data provides a novel multi-dimensional map of cancer cell lines that can reflect the extent to pan-cancer cell lines represent primary tumors, and an approach to evaluate the importance of omic features in cancer classification.

# INTRODUCTION

Disease classification provides key foundations for identification and treatment of diseases, especially for complicated diseases such as cancer (*Song, Merajver & Li, 2015*). Traditional classification of cancer was based on diseased organ, shared clinical symptoms and histological type (*Dozmorov, 2018*; *Ogino, Fuchs & Giovannucci, 2012*; *Song, Merajver & Li, 2015*). In recent years, the rapid development of high-throughput omics techniques and the accumulation of omics data enhances deeper understanding of cancer classification and

Corresponding authors
Song He, hes1224@163.com
Xiaochen Bo, boxc@bmi.ac.cn

characterization in oncology research (*Song, Merajver & Li, 2015*). Molecular classification based on omics data is now becoming important evidence for individual treatment of several cancer subtypes (*Koboldt et al., 2012*; *Tan et al., 2019*). Several studies have systematically characterized molecular classification of multiple cancer types based on The Cancer Genome Atlas (TCGA) and other projects (*Heim et al., 2014*; *International Cancer Genome C et al., 2010*). As early as 2014, *Hoadley et al. (2014)* reported an integrated analysis of 12 different cancers across six platforms, and redefined cancer types based on molecular characteristics. In 2018, the group further identified 28 distinct molecular pan-cancer subtypes arising from 33 cancers by integrating four types of omics data, providing a supplementing classification system to anatomic taxonomy (*Hoadley et al., 2018*).

Due to the difficulties to collect clinical samples from cancer patients, cancer cell lines had been still widely used as in vitro model for exploring cancer occurrence, development, and treatments. In some cancer types, the classification analysis of cancer cell lines has been proved to be a convenient way to characterize cancer sample subtypes. For example, five subtypes of colorectal cancer were revealed by iterative clustering of 74 different colorectal cancer cell lines, reflecting the consistency with the clinical classification of colorectal cancer patients (*Schlicker et al., 2012*). On the other hand, however, the reliability of cell lines as in vitro models of cancer samples has been doubted repeatedly. Previous studies have shown that in some cancer types, existing cell lines do not fully represent all tumor subtypes. For example, *Domcke et al. (2013)* compared the similarities and differences between the high-grade serous ovarian carcinomas cell lines and the primary tumors that they represent. They found that the most representative ovarian carcinomas cell lines were rarely studied as in vitro models, while other ovarian cancer cell lines were commonly used (*Domcke et al., 2013*). In addition, there is still lack of a comprehensive and complete profiling of pan-cancer cell lines classification based on integrated multiple omics data. Recently, *Li et al. (2017)* used reverse phase protein arrays data to divide about 650 cell lines into 10 pan-cancer groups which contributed a molecular portrait of cancer cell lines based on proteomics. Although great progresses have been made in the classification of pan-cancer samples based on integrated multiple omics data, there has been no research for providing an integrated molecular view of cancer cell lines based on multiple omics data and few publications have attempted to compare the classification of pan-cancer cell lines and patient samples.

In this study, we presented a systematic study on pan-cancer cell line classification based on single and integrated multiple omics data from the Cancer Cell Line Encyclopedia (CCLE) database (*Ghandi et al., 2019*). Our study seeks to provide a molecular classification to show a novel multi-dimensional map of pan-cancer cell lines and to compare the classification results obtained from our analysis with those of patient samples. The pan-cancer cell lines from CCLE were clustered in terms of single and multiple omics data using mRNA sequence data (mRNA), miRNA expression data (miRNA), copy number variation data (CNV), DNA methylation data (METHY) and reverse-phase protein array data (RPPA). Distinct molecular groups were identified by integrating five omics data. By characterizing each group by functional and cell-of-origin enrichment analysis, we confirmed significant molecular heterogeneity even among different cell lines of the same

cancer type. Several pan-organ system clusters and a pan-squamous morphology carcinoma cluster were also found among these molecular groups. By comparing with single omics clustering, we found that integrated multiple omics clustering could significantly capture more information of omics data. Additionally, we quantified the contribution analysis of each omics data for integrated clustering. The comparison of patient samples and cell lines classification results reveals that the classification of patient samples are more diverse and abundant than cancer cell lines.

## MATERIALS & METHODS

### Cancer cell lines and data pre-processing

Our study involved 1,019 cell lines from 31 previously established cancer types. The mRNA, miRNA, CNV, METHY and RPPA data were downloaded from the CCLE database for all cell lines (https://portals.broadinstitute.org/ccle/data) (*Ghandi et al., 2019*). The number of cancer cell lines and the cancer types involved were shown in Table 1.

For mRNA sequence data, we used RSEM values in gene level shared by CCLE database. We used miRNA expression data from CCLE for miRNA analysis. For DNA methylation, the promoter CpG data was used for clustering analysis. And reverse phase protein array data was downloaded for protein analysis. In parallel, we downloaded segmented copy number profiles from CCLE database for CNV analysis. This SNP6.0 arrays data was used as the input data for Gistic2.0 software (*Mermel et al., 2011*). Before pre-processing the data, we mapped segmented copy number to the chromosome arm level using Gistic2.0. This copy number variation by the chromosome arm level was the input data of CNV clustering analysis. Next, the following steps were performed to improve the dataset quality for single omics clustering.

(1) For each omics dataset, cell lines with more than 20% features missing, and features with more than 20% cell lines missing were filtered out.
(2) For each omics dataset, the missing data points were filled in using average imputations.
(3) For mRNA and miRNA data, log2 $(x+1)$ ($x$ is the value of mRNA and miRNA) transformation were performed before feature selection.
(4) For mRNA and METHY data, only features in the top 5,000 in terms of variance were selected. For miRNA, RPPA and CNV data, all features were considered.

For integrated multiple omics clustering, 670 cell lines from 24 cancer types with complete five omics data were used after samples alignment and deletion of cancer types with few number of cell lines.

### Single and multiple omics clustering of cell lines

For single omics dataset, we performed hierarchical clustering with different methods and distance measurements (Table 2). We used 30 clustering validity indices to select the optimal clustering number using the R package "NbClust" (version 3.0) (*Charrad et al., 2014*). The optimal parameters for function "NbClust" were set as follows: min.nc = 10, max.nc = 30, method = "average".

The Similarity Network Fusion and Consensus clustering algorithm (SNF-CC), a method combining Similarity Network Fusion (SNF) and Consensus clustering (CC) together to

**Table 1 The number of cancer cell lines of each type of omics data.**

| Tumor system | Cancer type | Abbreviation | Number of cell lines of mRNA data | Number of cell lines of miRNA data | Number of cell lines of CNV data | Number of cell lines of METHY data | Number of cell lines of RPPA data |
|---|---|---|---|---|---|---|---|
| | Acute lymphoblastic leukemia | ALL | 31 | 31 | 32 | 30 | 29 |
| | Chronic lymphoblastic leukemia | CLL | 4 | 4 | 4 | 2 | 4 |
| Hematopoietic lymphatic malignancies | Lymphoid neoplasm diffuse large B-cell lymphoma | DLBC | 39 | 39 | 40 | 37 | 37 |
| | Acute myeloid leukemia | LAML | 35 | 31 | 36 | 28 | 31 |
| | Chronic myeloid leukemia | LCML | 14 | 14 | 15 | 11 | 13 |
| | Multiple myeloma | MM | 28 | 28 | 30 | 19 | 27 |
| | Bladder urothelial carcinoma | BLCA | 25 | 25 | 23 | 19 | 24 |
| Urologic system malignancies | Kidney renal clear cell carcinoma | KIRC | 33 | 23 | 36 | 21 | 21 |
| | Prostate adenocarcinoma | PARD | 8 | 7 | 8 | 6 | 7 |
| | Breast invasive carcinoma | BRCA | 51 | 50 | 53 | 46 | 47 |
| Gynecologic cancers | Cervical squamous cell carcinoma and endocervical adenocarcinoma | CESC | 2 | 0 | 0 | 0 | 0 |
| | Ovarian carcinoma | OV | 47 | 49 | 49 | 43 | 47 |
| | Uterine corpus endometrial carcinoma | UCEC | 28 | 28 | 28 | 22 | 28 |
| | Cholangiocarcinoma | CHOL | 8 | 8 | 0 | 7 | 8 |
| | Colon adenocarcinoma/rectum adenocarcinoma | COAD/READ | 59 | 58 | 57 | 52 | 57 |
| Digestive system tumors | Esophageal carcinoma | ESCA | 27 | 25 | 27 | 17 | 26 |
| | Liver hepatocellular carcinoma | LIHC | 25 | 25 | 26 | 19 | 23 |
| | Pancreatic adenocarcinoma | PAAD | 41 | 40 | 43 | 35 | 40 |
| | Stomach adenocarcinoma | STAD | 37 | 37 | 38 | 32 | 37 |
| | Glioblastoma multiforme | GBM | 33 | 33 | 34 | 30 | 34 |
| Nervous system tumors | Brain lower grade glioma | LGG | 10 | 9 | 13 | 6 | 9 |
| | Medulloblastoma | MB | 4 | 4 | 4 | 4 | 4 |
| | Neuroblastoma | NB | 16 | 16 | 17 | 14 | 16 |
| | Lung adenocarcinoma | LUAD | 76 | 73 | 70 | 62 | 74 |
| Thoracic tumors | Lung squamous cell carcinoma | LUSC | 26 | 25 | 23 | 15 | 21 |
| | Mesothelioma | MESO | 9 | 9 | 9 | 7 | 8 |
| | Small cell lung cancer | SCLC | 50 | 50 | 53 | 47 | 45 |

Table 1 (*continued*)

| Tumor system | Cancer type | Abbreviation | Number of cell lines of mRNA data | Number of cell lines of miRNA data | Number of cell lines of CNV data | Number of cell lines of METHY data | Number of cell lines of RPPA data |
|---|---|---|---|---|---|---|---|
| | Head and neck squamous cell carcinoma | HNSC | 33 | 33 | 32 | 31 | 32 |
| Others | Sarcoma | SARC | 37 | 38 | 31 | 31 | 37 |
| | Skin cutaneous melanoma | SKCM | 54 | 55 | 54 | 51 | 55 |
| | Thyroid carcinoma | THCA | 11 | 12 | 12 | 11 | 12 |

**Table 2  Methods and measurements for hierarchical clustering.**

| Omics data | Method | Distance measurement |
|---|---|---|
| mRNA | ward.D | 1-Pearson's correlation coefficient |
| miRNA | ward.D2 | 1-Pearson's correlation coefficient |
| CNV | ward.D2 | Manhattan Distance |
| METHY | ward.D | 1-Pearson's correlation coefficient |
| RPPA | ward.D2 | 1-Pearson's correlation coefficient |

take advantage of both for cancer type identification, was applied to integrate multiple omics data (*Monti et al., 2003*; *Wang et al., 2014*). The method was implemented by the R package "CancerSubtypes" (version 1.8.0) (*Xu et al., 2017*). The optimal parameters for function "ExecuteSNF.CC" from "CancerSubtypes" were set as follows: $K = 20$, alpha $= 0.5$, $t = 20$, maxK $= 30$, pItem $= 0.8$, reps $= 500$. The silhouette coefficient, a measurement of consistency of each object within clusters, also derived using the function "silhouette_SimilarityMatrix" from the R package "CancerSubtypes".

The function "pamr.listgenes" from the "pamr" R package (version 1.56.1) was used to find the most suitable clustering features for the illustration of clustering by heatmap (*Tibshirani et al., 2002*).

## Dominant cancer type and functional enrichment analysis

A hypergeometric distribution was used to calculate the *P*-value for cancer types in each cluster. Cancer types with a *P*-value $<10^{-3}$ were chosen as the dominant types of each cluster. The $-\lg(P\text{-value})$ represents the enrichment score that cancer types gathered in each cluster.

We explored the differential expressed genes (DEGs) among clusters using the R package "limma" (version 3.38.3) (*Ritchie et al., 2015*). Genes with adjusted *P*-value $<0.05$ were selected, and were further screened according to |log2 fold-change|$>1$. Finally, the above DEGs were fed into enrichment analysis with GO and KEGG terms using the R package "clusterProfiler" (version 3.10.1) (*Yu et al., 2012*). The significantly enriched pathways were identified using false discovery rate $<0.05$.

## Feature contribution of integrated multiple omics clustering

We used the normalized mutual information (NMI), which was a measure of the interdependence between two random variables, to measure the contribution of each omics type feature. The function "rankFeaturesByNMI" in the R package "SNFtool"

(version 2.3.0) were used to compute NMI (*Liu et al., 2018a*; *Wang et al., 2014*). Codes are provided in Script S1.

## Tumor maps of cancer cell lines

We used the TumorMap website to create pan-cancer cell lines maps from the above integrated data. TumorMap is an interactive website for assisting in exploring high-dimensional and complicated omics data (https://tumormap.ucsc.edu/) (*Newton et al., 2017*). In TumorMap, samples are distributed on a hexagonal grid based on their similarity and rendered using Google's Map technology. The distances were used as input to generate a 2D layout of the samples. We used features that NMI ranks top 20% to calculate Euclidean similarity between each cell line. The Euclidean similarity is equal to $1/(1 +$ Euclidean distance). All parameters in TumorMap were set default.

# RESULTS

## Clustering based on single omics data

We initially clustered cell lines based on each type of omics data, which were mRNA, miRNA, CNV, METHY and RPPA data. The optimal clustering numbers were set to 10 (Fig. 1 and Fig. S1).

In the hierarchical clustering result of 901 cell lines by mRNA (Fig. 1A, Table S1 and Fig. S1A), we found that one cluster was mainly formed from a single type of cancer (C7 [SKCM]). Additionally, hematopoietic lymphatic malignancies were separated into two clusters, (C6 [ALL-DLBC-MM] and C9 [LAML-LCML]). Cancer cell lines with histological similarity or proximity tended to group together. These include C2: pan-gastrointestinal [COAD/READ-STAD], C4: nervous system tumors [GBM-LGG and some SARC whose features were the same as them] and C8: pan-gynecological [OV-UCEC and other SARC cell lines]. KEGG enrichment analysis indicated that C1 was enriched in human cytomegalovirus infection, transcriptional misregulation in cancer, proteoglycans in cancer, TNF signaling pathway and NF-kappa B signaling pathway (Figs. S2A and S2B). Meanwhile, the cell lines in C1 were enriched in GO terms including reproductive system development and morphogenesis of embryonic epithelium (Figs. S2C–S2H).

In the clustering result of miRNA data of 879 cell lines (Fig. 1B, Table S2 and Fig. S1B), five clusters predominately contained a single cancer type (C2 [COAD/READ], C5 [STAD], C6 [HNSC], C8 [SKCM] and C10 [NB]). And tumors of the hematopoietic lymphatic system were distributed in two clusters (C4 [ALL-DLBC-LAML-LCML-MM] and C9 [ALL-DLBC-LAML]). The significant signature of these two clusters were high expression of has-miR-142-5p and has-miR-142-3p, which played an important role in lineage differentiation of hematopoietic cells (*Sharma, 2017*).

CNV data sorted at the chromosome arm-level for 897 cell lines were divided into 10 clusters through hierarchical clustering, four clusters mainly formed from a single cancer type (C3 [GBM], C5 [SKCM], C7 [PAAD] and C8 [SCLC]) (Fig. 1C, Table S3 and Fig. S1C). C8 was characterized by the deletion of chr3p and chr17p and the amplification of chr3q. This characterization had been reported in previous studies (*Carter et al., 2017*; *George et al., 2015*; *Peifer et al., 2012*). C1 and C10 were enriched for ALL, DLBC, LAML

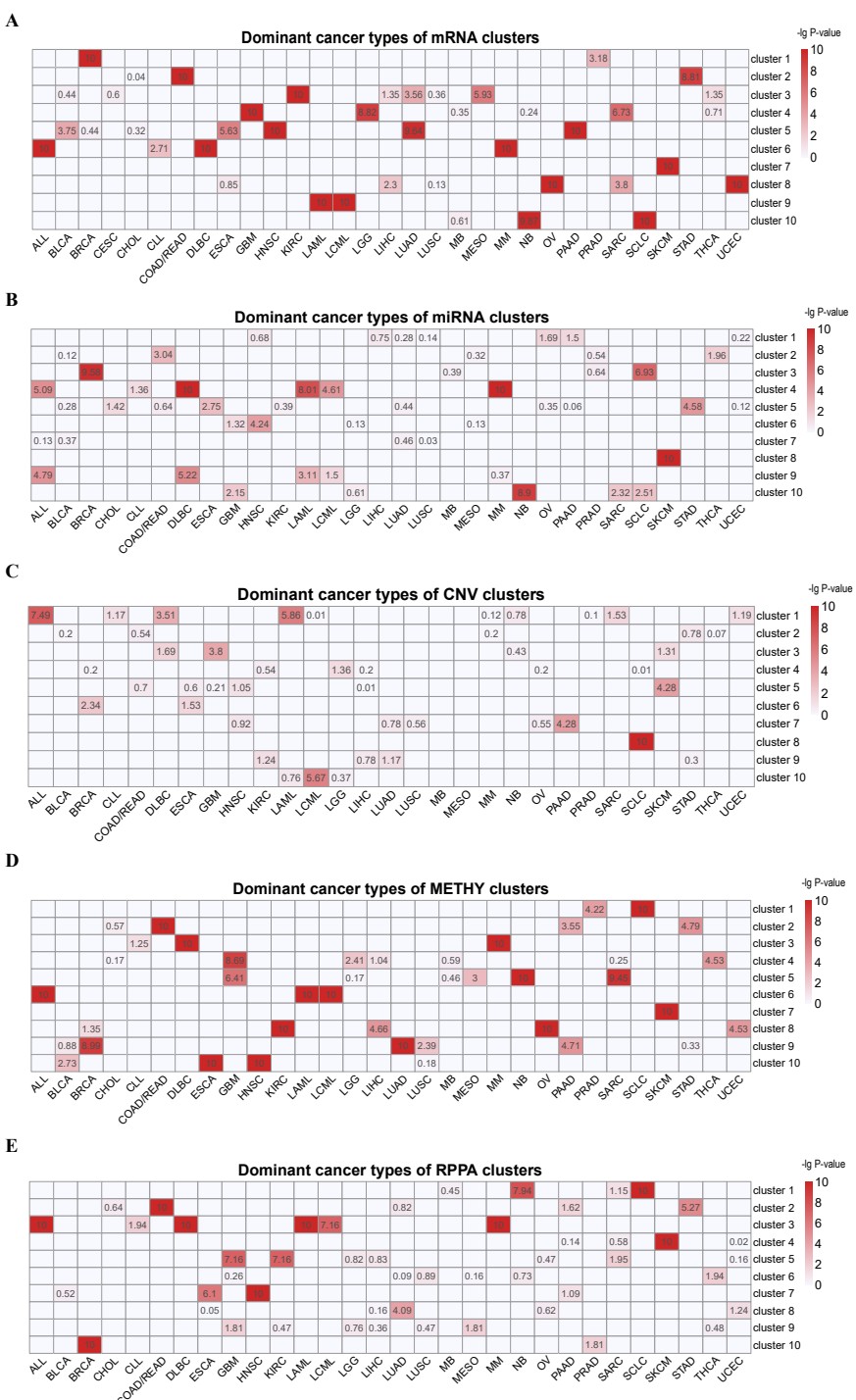

**Figure 1** **Cluster labels of single omics clustering.** (A) mRNA. (B) miRNA. (C) CNV. (D) METHY. (E) RPPA. A hypergeometric distribution was used to calculate the *P*-value for cancer types in each cluster. The rows represent clusters, and the columns represent cancer types. The values represent the $-\lg(P\text{-}value)$ of cancer types. Cancer types with $-\lg(P\text{-}value) > 3$ in each cluster were defined as dominant cancer types. All the blank cells mean the instances of *P*-value = 0.

and LCML. We observed fewer alterations in C1 but more alterations in C10. For example, C10 was characterized by chr8, chr19 and chr6 copy number increase.

Among the unsupervised clustering result of 755 cell lines using METHY data (Fig. 1D, Table S4 and Fig. S1D), there was one cluster that virtually consisted of one cancer type (C7 [SKCM]). Meanwhile, hematopoietic lymphatic malignancies were still enriched in two clusters (C3 [DLBC-MM] and C6 [ALL-LAML-LCML]). Cancer cell lines originating from same organ often gathered in the same cluster, such as C2 [COAD/READ-STAD-PAAD], a group of digestive system cancers whose common features were the high expression of CNKSR1, FOLH1, ADGRG1, SMAD7, LRATD1 and MVP (*Haffner et al., 2009*; *Ji et al., 2018*; *Kobayashi et al., 2006*; *Quadri et al., 2017*; *Slattery et al., 2010*; *Teng et al., 2017*). Additionally, squamous morphology cancer cell lines aggregated by METHY patterns (C10 [ESCA-HNSC]), particularly in terms of ARHGDIB and SEPTIN9 loss (*Bennett et al., 2008*).

In hierarchical clustering of RPPA data from 854 cell lines (Fig. 1E, Table S5 and Fig. S1E), C4 [SKCM], C8[LUAD] and C10 [BRCA] mostly contained one cancer type. The characteristics of C10 in this analysis had high level of ER, GATA3, AR, ERBB2, FASN, PREX1, CDH1 and CLDN7, and low level of CAV1 (*Barrio-Real et al., 2016*; *Neve et al., 2006*; *Taherian-Fard, Srihari & Ragan, 2015*). Hematopoietic lymphatic malignancies were enriched in one cluster (C3 [ALL-DLBC-LAML-LCML-MM]). Consistent with METHY analysis, a pan-gastrointestinal carcinoma cluster with COAD/READ-STAD was gathered in C2, which had high level of CDH1, CLDN7 and TYRO3, and a low level of CAV1 (*Burgermeister et al., 2007*; *Di Bartolomeo et al., 2016*; *Qin & Qian, 2018*). In addition, C7 was an enrichment cluster of squamous morphology cancer cell lines, mostly made of HNSC and ESCA and was characterized by high level of CDH1, CLDN7 and CAV1 (*Ando et al., 2007*; *Bello et al., 2008*; *Shah et al., 2009*).

Interestingly, some cancer types, such as SKCM, were individually classified in all five omics data, whereas some cancer types such as SCLC and BRCA were clustered individually only in one or two omics data. This result indicates that exposed information of each omics data is different at molecular level. According to investigation, SKCM is a paradigm of invasive cancer characterized by the highest mutational frequency among all cancer types and a large accumulation of changes in transcriptome (*Cancer Genome Atlas N, 2015*; *Lawrence et al., 2013*). Compared to other cancer cell lines, there are a large number of special molecular characteristics in SKCM cell lines. For example, we found that the levels of miR-188-3p and miR-514 were increased significantly, whereas in other cell lines, the levels of the two miRNA were decreased. At the CNV level, amplification of chr7 was found in most SKCM cell lines. It is generally known that several common mutations of SKCM, are on the chromosome 7 (*Hayward et al., 2017*). Moreover, pan-cancer clusters could be found based on mRNA, METHY and RPPA, but were not individually clustered in miRNA and CNV. This phenomenon was consistent with feature contributions of integrated multiple omics clustering, and was related to the fact that the characteristic information from mRNA, METHY and RPPA dataset were more representative.

## Integrated clustering based on multiple omics data

By using SNF-CC, we integrated all five of omics datasets (mRNA, miRNA, CNV, METHY and RPPA) across 670 cell lines and identified 18 clusters (Fig. 2A, Table S6 and Fig. S3).

For these 18 clusters, 12 of them were dominated by a single cancer type (C1 [SCLC], C2[GBM], C4 [ALL], C6 [SARC], C7 [BRCA], C8 [ALL], C10 [MM], C12 [DLBC], C14 [LAML], C15 [SKCM], C16 [NB] and C17 [KIRC]) (Figs. 2A and 2B, Fig. S4). And each clusters also mixed with few amounts of other cancer types. Except SKCM, C15 also contained one glioblastoma multiforme cell line (LN229) with low level of VHL and high expression of has-miR-146a, has-miR-29b and has-miR-188-3p (*Aurich, Fleming & Thiele, 2017*). It is notable that although SARC is the dominant cancer types in C6, the proportion within the cluster is relatively low.

There were six clusters that dominated by two cancer types (C3 [PAAD-LUAD], C5 [HNSC-ESCA], C9 [LAML-LCML], C11 [COAD/READ-STAD], C13 [LUAD-LIHC] and C18 [OV-UCEC]) (Figs. 2A and 2B, Fig. S4). On the one hand, the proportion of two dominant cancer types were almost equal in C3, C13 and C18. And C3 was characterized by high level of CDH1. On the other hand, in C5, C9 and C11, one of the two dominant cancer type was over 50%. And C9 had high levels of VAV1 and STAT5A and low level of CTNNB1 (*Bertagnolo et al., 2011*; *Harir et al., 2007*; *Ysebaert et al., 2006*).

Three pan-cancer clusters influenced by organ of origin or cell morphology pattern were obtained. These clusters included pan-gastrointestinal cluster (C11 [COAD/READ-STAD]), pan-gynecological cluster (C18 [OV-UCEC]) and pan-squamous morphology carcinoma cluster (C5 [HNSC-ESCA]). For pan-gastrointestinal cluster (C11 [COAD/READ-STAD]), KEGG enrichment analysis results showed that these cell lines shared down-regulated in cytokine-cytokine receptor interaction and TNF signaling pathway (Figs. 3A and 3B and File S1). And cell lines in C11 had high levels of protein binding involved in heterotypic cell–cell adhesion, SNARE binding and G-protein beta-subunit binding in GO terms (Figs. 3C–3H and File S1). Meanwhile, pan-squamous morphology carcinoma cell lines (C5 [HNSC-ESCA]) were characterized by up-regulated of nicotine addiction and cell adhesion molecules pathway. And these cell lines had high levels of CAV1, EGFR and ITGA2 (*Ando et al., 2007*; *Song et al., 2015*).

We also observed two clusters with the same cancer type dispersed. For instance, cell lines from ALL were divided into two clusters, C4 and C8, despite the common characteristics such as Human T-cell leukemia virus 1 infection, Th17 cell differentiation, and TNF signaling pathway. The ALL cell lines in C8 were enriched in KEGG terms including up-regulated in ECM-receptor interaction, down-regulated in antigen processing and presentation pathway, while the ALL cell lines in C4 had low level of cellular senescence (Figs. 3A, 3B and File S1). GO enrichment analysis results showed that the ALL cell lines in C8 had low level of cell–cell junction and high levels of calcium channel activity, while the ALL cell lines in C4 were down-regulated in growth factor receptor binding and sulfur compound binding (Figs. 3C–3H and File S1). At other four omics levels, the features of these two clusters were different as well. For example, the levels of PTEN (a tumor suppressor gene), LCK and Syk (two immune-related genes) and has-miR-151-5p (related to tumor invasion and metastasis) was completely inconsistent.

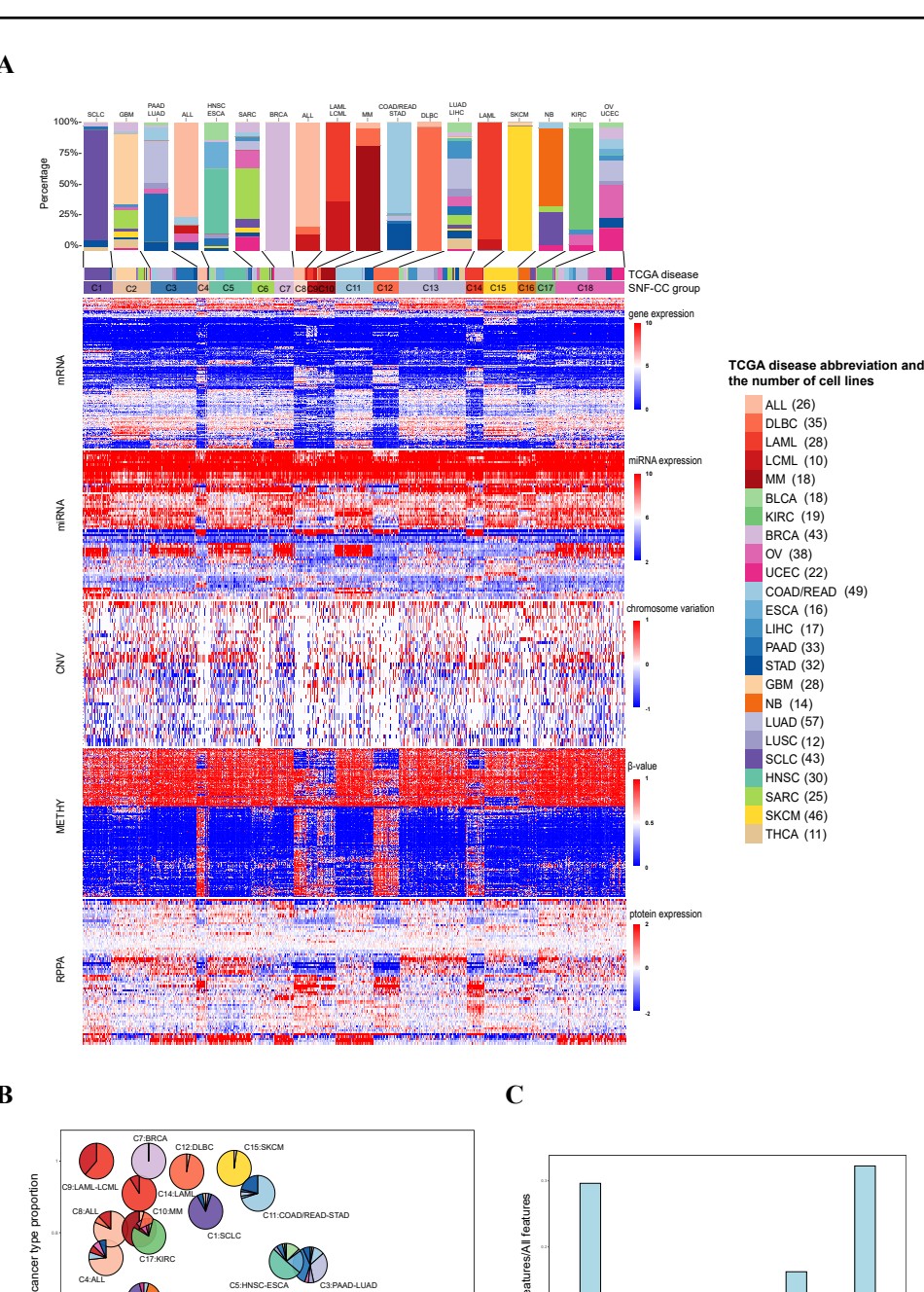

**Figure 2 Classification of pan-cancer cell lines based on integrated multiple omics data.** (A) Data integrated analysis of SNF-CC. Types of cancer cell line are color-coded as shown in the right. The first track represents cell lines of TCGA disease. The second track represents the SNF-CC group. A bar graph was used to show cancer types and the proportion of cell lines in each cluster. The dominant cancer types of each cluster were marked on the top of the bar graph. (B) Clusters composition. Pie charts show the cancer type composition within clusters and the proportion of the membership. The $y$-coordinate of each pie center reflected the dominant cancer types proportion. The $x$-coordinate was determined by the number of cell lines in each cluster. (C) The contributions of feature (top 20% NMI) from each omics data.

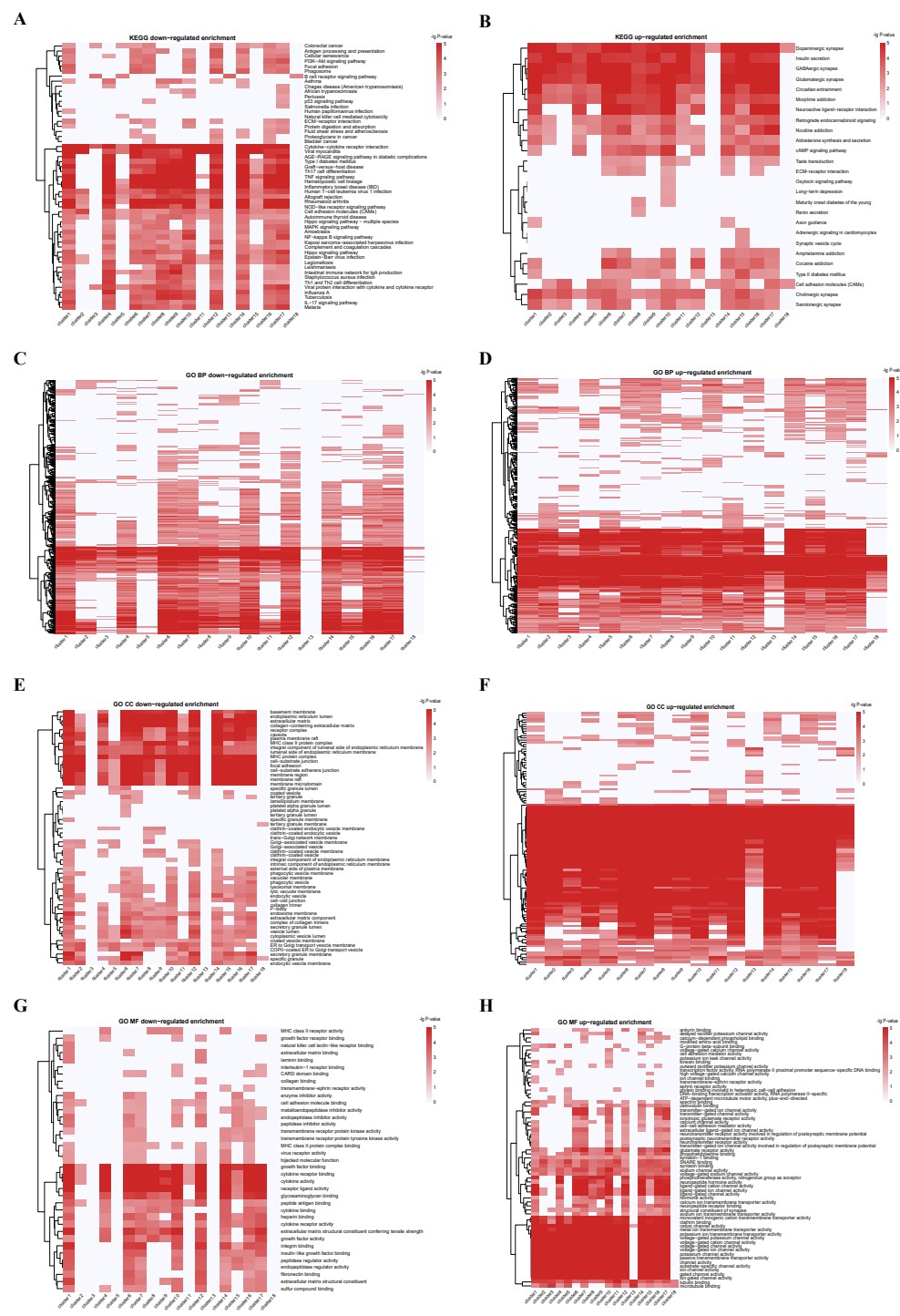

**Figure 3** **KEGG and GO enrichment analyses from integrated multiple omics clustering.** (A) KEGG enrichment heatmap of down-regulated genes. (B) KEGG enrichment heatmap of up-regulated genes. (C) GO biological process enrichment heatmap of down-regulated genes. (D) GO biological process enrichment heatmap of up-regulated genes. (E) GO cellular component enrichment heatmap of down-regulated genes. (F) GO cellular component enrichment heatmap of up-regulated genes. (G) GO molecular function enrichment heatmap of down-regulated genes. (H) GO molecular function enrichment heatmap of up-regulated genes. Deeper red color signifies greater enrichment score in A–H.

**Table 3  The percentages of the top 20% NMI features from each omics data.**

|  | mRNA | miRNA | CNV | METHY | RPPA |
|---|---|---|---|---|---|
| The top 20% NMI features | 5626 | 67 | 1 | 7691 | 69 |
| All features | 18,996 | 654 | 40 | 47,362 | 214 |
| Percentage | 29.62% | 10.24% | 2.50% | 16.24% | 32.24% |

Integrated multiple omics clustering provided a global view of cancer types because it could capture both shared and complementary information from each omics data. Several cancer types which mixed together in one single omics data were divided in other single omics data or integrated omics data. For example, BRCA and SCLC were mixed together based on miRNA data, but they were separated into two distinct molecular clusters based on integrated omics data. Besides, in single omics clustering, three pan-organ system clusters were only found based on mRNA data and the pan-squamous morphology carcinoma cluster was only found based on METHY and RPPA data. But pan-gastrointestinal cluster, pan-gynecological cluster and pan-squamous morphology carcinoma clusters were simultaneously identified by integrated multiple omics clustering.

The relative contribution of each omics data to the integrated clustering was computed based on the NMI value. On the basis of the top 20% statistical features from the five omics data, we found that RPPA and mRNA contributed 32.24% and 29.62% respectively, followed by METHY (16.24%) (Fig. 2C and Table 3). This result demonstrated that mRNA and proteomics data were particular important for cancer molecular classification. Meanwhile, more information was showed based on mRNA and RPPA data than other omics data in single omics clustering. For instance, pan-organ system clusters were identified based on mRNA and RPPA data, but not in miRNA and CNV. This results indicated that mRNA and proteomics data could be preferred if multiple omics data were not able to be measured simultaneously.

## The comparison of classification between cancer samples and cell lines

We compared the classification results of 19 cancer types shared by cancer cell lines from CCLE and patient samples from TCGA (*Hoadley et al., 2018*). Clusters of patient samples and cell lines were divided into three types respectively, namely clusters dominated by single cancer type, pan-cancer clusters and clusters mixed with other cancer types (Table 4).

For hematopoietic lymphatic malignancies, the classification of cancer cell lines is more abundant than patient samples. For example, some LAML cell lines were clustered together in a group, while others were mixed with LCML cell lines into another group (LAML-LCML) in our findings. For patient samples, there is only one LAML group (*Hoadley et al., 2018*). The classification of DLBC cell lines was consistent with patient samples. Just like hematopoietic malignancies, the SARC patient samples were clustered individually into a group. However, for cell lines, except gathering in a single group, a few other SARC cell lines were mixed with GBM.

For most solid tumors, the classification of patient samples is generally more abundant and diverse than the corresponding cell lines. In general, patient samples with same cancer

**Table 4  The comparison of classification between cancer samples and cell lines.**

| Cancer types | Classification of cancer samples in TCGA | | | Classification of cell lines in CCLE | | |
|---|---|---|---|---|---|---|
| | Number of clusters dominated by single cancer type | Number of pan-cancer clusters | Number of clusters mixed with other cancer types | Number of clusters dominated by single cancer type | Number of pan-cancer clusters | Number of clusters mixed with other cancer types |
| DLBC | 1 | 0 | 0 | 1 | 0 | 0 |
| LAML | 1 | 0 | 0 | 1 | 0 | 1 |
| BLCA | 0 | 0 | 1 | 0 | 0 | 1 |
| KIRC | 0 | 1 | 0 | 1 | 0 | 0 |
| BRCA | 3 | 0 | 1 | 1 | 0 | 0 |
| OV | 1 | 0 | 0 | 0 | 1 | 0 |
| UCEC | 1 | 0 | 0 | 0 | 1 | 0 |
| COAD/READ | 0 | 2 | 0 | 0 | 1 | 0 |
| ESCA | 0 | 2 | 0 | 0 | 1 | 0 |
| LIHC | 1 | 0 | 0 | 0 | 0 | 1 |
| PAAD | 0 | 0 | 1 | 0 | 0 | 1 |
| STAD | 1 | 1 | 1 | 0 | 1 | 0 |
| GBM | 0 | 1 | 0 | 1 | 0 | 0 |
| LUAD | 1 | 0 | 0 | 0 | 0 | 2 |
| LUSC | 0 | 1 | 0 | 0 | 0 | 1 |
| HNSC | 0 | 2 | 0 | 0 | 1 | 0 |
| SARC | 0 | 0 | 1 | 1 | 0 | 1 |
| SKCM | 0 | 0 | 1 | 1 | 0 | 0 |
| THCA | 1 | 0 | 0 | 0 | 0 | 1 |

type can be divided into multiple groups, while cell lines with same cancer type are clustered in one group. For example, the samples of breast cancer were classified into three subgroups (chr8q amp, HER2 amp and Luminal). In addition, there were a large number of BRCA samples gathered with other cancer types in a mixed cluster. Except a few of BRCA cell lines were mixed in a pan-gynecological cluster, whereas almost all BRCA cell lines were clustered in a single group. And there are similar clustering results in gastrointestinal cancer and squamous cell carcinoma. There were two pan-gastrointestinal groups and a single gastric cancer group in patient samples (*Hoadley et al., 2018*). While most colorectal and gastric cancer cell lines were clustered in a pan-gastrointestinal cluster without forming a STAD group in the classification result of cell lines. Most patient samples of ESCA were divided into a pan-squamous morphology carcinoma group and a pan-gastrointestinal group, while in our study, most ESCA and HNSC cell lines were clustered in a pan-squamous morphology carcinoma cluster. This indicates that not all molecular subtypes of patient samples can be represented by current panels of cancer cell lines.

On the other hand, while the number of clustering groups were same, the types of clusters were different in cell lines and patient samples of some cancer types. For example, there is a pan-kidney group in the classification of patient samples. But this cluster was not in our results because there is only one cell line associated with kidney cancer, renal clear cell carcinoma (KIRC), in the CCLE database (*Hoadley et al., 2018*; *Ricketts et al.,*

*2018*). Although there was no pan-kidney cluster in our research, the KIRC cell lines were clustered individually in our clustering results. And the patient samples of OV and UCEC were gathered in different clusters, while cell lines of the two cancer types were clustered together in a pan-gynecological group. GBM patient samples were clustered with LGG to form a pan-cancer group, however due to lacking LGG cell lines, GBM cell lines were clustered individually. The classification of patient samples in some cancer types, such as LIHC, LUSC and THCA, the clusters they formed were dominated by single cancer type (*Hoadley et al., 2018*). However, the clusters formed by these three types of cancer cell lines were mixed with large number of other cancer types.

In general, due to the larger numbers of patients and the greater heterogeneity in TCGA samples, the classification results of patient samples are more diverse and abundant than cancer cell lines. It is clear that the existing cancer cell lines do not fully represent all the molecular types of corresponding patient samples. However, in some cancer types, the classification of patient samples and cell lines were consistent. This shows that cancer cell lines can represent primary samples to some extent.

### The TumorMap landscape of pan-cancer cell lines

We used TumorMap web tool to visualize the landscape of pan-cancer cell lines. The same layout and four different color schemes (SNF-CC cluster, TCGA disease, Pan-organ system and histology) were used to reveal that most cancer cell lines gathered based on organ systems and histopathological similarity (Fig. 4). More nuance within a cancer type were apparent. The SARC was widely distributed and separated into three parts. Most of them enriched in C6 (SARC), while others were fell into C2 (GBM) characterized by amplification of chr7p and C16 (NB) characterized by chr17q amplification (Figs. 4A and 4B). Additionally, major cell lines from STAD were assembled in C11 (COAD/READ-STAD), but few gathered in C18 (OV-UCEC) and C13 (LUAD-LIHC). Pan-organ system clusters and pan-squamous morphology carcinoma cluster reported previously were shown on the map (Fig. 4C) (*Berger et al., 2018*; *Campbell et al., 2018*; *Liu et al., 2018b*). We found that cell lines within C11 (COAD/READ-STAD) and C5 (HNSC-ESCA) were tightly gathered, while cell lines within C18 (OV-UCEC) were relatively dispersed. The TumorMap landscape showed that cancer cell lines with similar histology characterization tended to get together, even though histological information were not used during calculating similarities (Fig. 4D). The hematopoietic lymphatic malignancies were remote from other cancer types on the map. This result underscored that the molecular characteristics of hematopoietic lymphatic malignancies were different from other cancer types (Fig. 4D). Moreover, C15 (SKCM) and C17 (KIRC) were also far away from other solid tumor groups on the map.

We downloaded the drug susceptibility data for 24 anticancer drugs across 504 cell lines in CCLE database. We used TumorMap web tool to analyze the relationship between the drug susceptibility and the pan-cancer clustering. We divided the analysis results into four types and chose the representative drugs as examples (Fig. S5).

(1)   These anticancer drugs have a strong effect on almost all cancer cell lines. For example, LUAD-LIHC (C13), GBM (C2), SARC (C6), DLBC (C12), SKCM (C15), OV-UCEC

(C18) and HNSC-ESCA (C5) cell lines are sensitive to Paclitaxel, a broad-spectrum anticancer drug (Fig. S5A).

(2) Almost all cell lines are not sensitive to these drugs, for example, L-685458, a gamma-secretase inhibitor (Fig. S5B).

(3) Only one cell line or few cell lines are sensitive to these anticancer drugs. For example, as a BRAF inhibitor, PLX4720 has an obvious effect on some SKCM (C15) cell lines, but has no effect on other cell lines (Fig. S5C).

(4) These anticancer drugs have a strong effect on many cancer cell lines, but have a weak effect on others. For example, RAF256 is a dual inhibitor of mutant BRAF and vascular endothelial growth factor receptor 2. This drug can inhibit proliferation of SKCM (C15), GBM (C2), LUAD-LIHC (C13), OV-UCEC (C18) and some COAD/READ-STAD (C11) cell lines (Fig. S5D).

## DISCUSSION

In this research, we provide a pan-cancer cell lines classification based on single and multiple omics data. First, unsupervised hierarchical clustering was performed using five omics data from CCLE database, involving 31 cancer types and more than 1,000 cell lines, with each omics data showing different characterization. Next, we analyzed integrated multiple omics data of pan-cancer cell lines using SNF-CC method and ultimately clustered 24 cancer types into 18 groups. Moreover, we analyzed dominant cancer types and functional enrichment of each clusters. Then, the relative contribution of each omics data were calculated. We compared the classification of cancer cell lines and patient samples. Finally, we used TumorMap web tool to illustrate the landscape of cancer cell line clusters.

Our study showed that clusters were strongly influenced by organ system and cell of origin. Three pan-cancer cell line clusters: pan-gastrointestinal group, pan-gynecological group and pan-squamous morphology carcinoma group were identified by integrated multiple omics clustering simultaneously (*Berger et al., 2018*; *Campbell et al., 2018*; *Liu et al., 2018b*). Common functional mechanism and multiple omics characterization in the same pan-cancer clusters may contribute to potential clinical application value. The clusters obtained by integrated clustering provided reference about treating the same disease with different therapies. On one hand, one cancer type with different molecular features gathered in different clusters. Although these cell lines belong to same cancer type, the treatment therapies based on molecular characterizations may be different. On the other hand, the treatment of a cluster containing multiple cancer types may be the same. The comprehensive analysis about cancer classification could be used to elucidate potential disease mechanism and provide additional guidance for molecular treatments.

Cancer cell lines have been commonly used as in vitro models of tumors in biomedical research, however, the reliability has been doubted. The comparison of molecular classification between cancer cell lines and patient samples from TCGA provided a valuable insight into the reliability of cell lines as samples. While the overall classification of cell lines and samples were quite similar, samples from cancer patients were generally more diverse and abundant than cell lines. In some types of cancer, the number of molecular

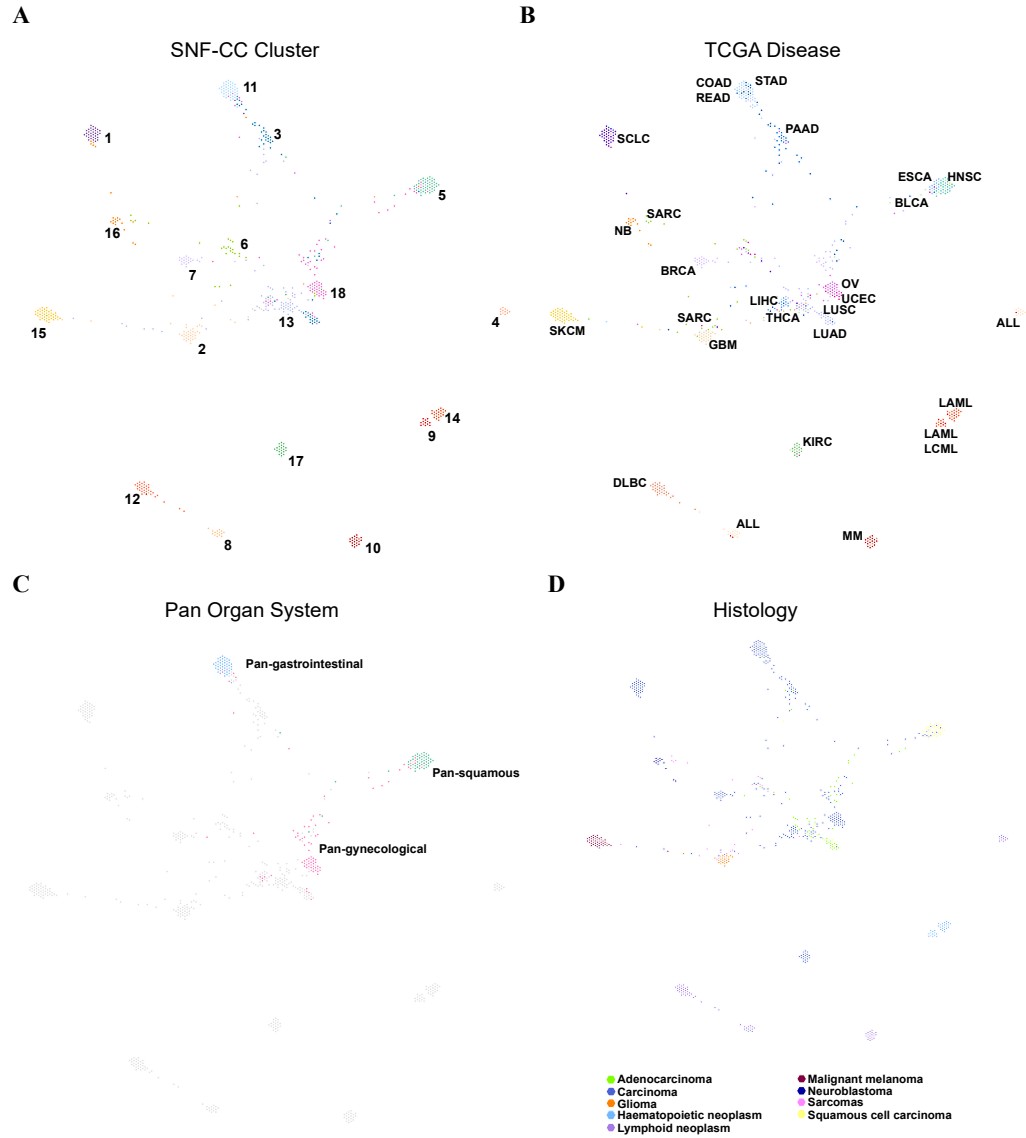

**Figure 4** **The SNF-CC TumorMap.** The TumorMap layout was computed from cell line Euclidean similarity by NMI features, and similar cell lines were adjacent to each other. Each node represents a single cell line and is colored with attributes including (A) SNF-CC cluster, (B) TCGA disease, (C) Pan organ system and (D) Histology.

groups in patient samples were more than the corresponding cancer types of cell lines, while in others, the molecular classification of patient samples matched the corresponding molecular groups of cell lines. Our study provides researchers with a widely comparison of pan-cancer cell lines and primary samples.

We also presented that mRNA and proteomics data were more strongly grouped in terms of classification by cancer type than other omics data. This is meaningful for biologists and oncologists choosing what types of omics data they need for their particular analysis.

## CONCLUSIONS

In summary, we provide a novel multi-dimensional landscape of cancer cell lines, and an approach to evaluate the importance of omic features in cancer classification. This research is crucial for treating same cancer with different therapies based on molecular characteristics. The comparison of molecular classification between pan-cancer cell lines and patient samples represents a valuable resource for the reliability of available cell lines as model of tumors. With the lower cost of omics analyses and the development of high-throughput omics technologies, integrated more omics data for cancer classification could be applied in clinical diagnosis and guide personalized treatment.

### Funding

This research was funded by the National Key R&D Program of China (2016YFC0901600). The funders had no role in study design, data collection and analysis, decision to publish, or preparation of the manuscript.

### Grant Disclosures

The following grant information was disclosed by the authors:
National Key R&D Program of China: 2016YFC0901600.

### Competing Interests

The authors declare there are no competing interests.

### Author Contributions

- Xiaoxi Yang conceived and designed the experiments, performed the experiments, analyzed the data, prepared figures and/or tables, authored or reviewed drafts of the paper, and approved the final draft.
- Yuqi Wen performed the experiments, prepared figures and/or tables, and approved the final draft.
- Xinyu Song analyzed the data, prepared figures and/or tables, and approved the final draft.
- Song He and Xiaochen Bo conceived and designed the experiments, authored or reviewed drafts of the paper, and approved the final draft.

### Data Availability

The raw data is available at the CCLE database for all cell lines (https://portals.broadinstitute.org/ccle/data) and these files are also available at Figshare: Yang, Xiaoxi (2020): Cancer cell lines raw data. figshare. Dataset. https://doi.org/10.6084/m9.figshare.12016968.v2.

The code is available as a Supplemental File.

## Supplemental Information

Supplemental information for this article can be found online at http://dx.doi.org/10.7717/peerj.9440#supplemental-information.

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
