# Peer review of "Exploring the classification of cancer cell lines from multiple omic views"

_PeerJ, doi:10.7717/peerj.9440_

## Round 0.1 · original submission · Major Revisions

The manuscript is principally interesting but all the issues raised by the referees must be addressed before it can be considered for publication. The points raised must be addressed by modifications of the analyses not by simply discussing them. Special attention should be devoted to a clear focus of the analysis.

Reviewer 1 ·

Basic reporting

In the manuscript entitled “Exploring the classification of cancer cell lines from multipleomics view”, Yang et al. addressed the multiomics-based, pancancer-scaled molecular taxonomy of cancer celllines. The study is based on the public, CCLE multiomics data encompassing mRNA/miRNA/SCNA/methylation/RPPA data. As clearly stated in Introduction, the value of multiomics datasets of cancer celllines has long been proposed and this study demonstrates the feasibility as well as some novel molecular taxons and interdependency among the datasets. I believe their results are of potential interests to general cancer biologists and clinicians, but I have a number of comments as the following.

Experimental design

No comments

Validity of the findings

No comments

Additional comments

Major comments:

1. Their results can be compared with similar work based on PanCancer-TCGA in terms of Fig.2B and Fig.2C. I believe that the most distinctive features of celllines compared to primary/TCGA tumors are high purity (i.e., the lack of intratumoral heterogeneity and contamination of normal tissues) and I am very curious how this features affect the PanCancer clustering.

2. One important features of CCLE database is that they include the data representing drug sensitivity. Fig. 4 can be further revised to demonstrate the relationship between the drug susceptibility or resistance and their pan-cancer clustering.

Minor comments
- It is hard to read the fonts in the main Figures, esp. Fig. 3 and also Fig. 4.
- In Abstract, “Results. 18 molecular clusters” should be revised into “Eighteen molecular clusters”.

Reviewer 2 ·

Basic reporting

Review Report
“Exploring the classification of cancer cell lines from multiple omic view”
Yang et al.

The authors classified single omics data including mRNA, miRNA, CNV, Methylation and Reverse Phase Protein Array from cell lines of the Cancer Cell Line Encyclopedia (CCLE) consortium and integrated multi omics data. Subsequently, the authors visualized the single and integrated data using the TumorMap tool to show the landscape of cancer cell lines based on molecular data.
Based on their analysis they identified 18 distinct clusters which consist of 3 pan-cancer clusters and one pan squamous cluster. Moreover, they showed that integrated omics analysis has an added value over single omics data analysis and that the mRNA and proteomics data contribute mostly to the distinct clusters.
Although there approach of integrated CCLE has added value I have some questions regarding the current manuscript.

Major comments.


• The description of the methods is very general and the results are likely very hard to reproduce based on the M&M section. Please elaborate more on the analysis methods and the parameter settings.
• The main message of the manuscript is not clear. Is it focused on the bioinformatics analyses methods or did the author wish to show new cancer biology? I strongly suggest to focus on the bioinformatics part, since the current added-value for cancer biology is limited as new testable cancer hypotheses are generated.
• Limma is used for differentially expression analysis, but this method is less suitable for Next Generation Sequencing data analysis. Why did the authors not use DESeq2 or EdgeR as analysis method. Please clarify.
• All the main figures and supplementary figures have very limited legend text. Therefore they are hard to impossible to interpret. In figure 1A the legend colors go from 0 -10. What unit is used? It seems that colors reflect just an on and off situation of a gene. Most dots are very blue for instance. So could a binary clustering better reflect the biology? Please clarify. Moreover, the first track is not clear. Maybe the authors could add a dendrogram to show the different groups more clearly. In addition the number of samples differ hugely between the groups. In 1A and 1B the dark purple TCGA group has about a 3-fold difference. Can the authors explain this difference in more detail by for instance add the number of samples in the legend text.
• Line 134 -190 discuss the clusters with labels such as C1 C9 etc. However, the reader cannot find these labels back in the manuscript. Therefore, the description of the results are very unclear.
• Please correct the clustering analysis for gender or show that the clusters are not a reflection of gender, since expression data is strongly biased to gender and might explain the pan-gynecological group.
• Line 197: The authors emphasize that the BRCA cluster C7 covers three prostate adenocarcinoma cell lines. This is a huge surprise since BRCA cell lines are mostly if not all female derived and prostate cancer cell lines are all male derived. Please show that this result is not an artifact of your analysis by selecting the features of the BRCA and the three prostate cancer cell lines that they share and give a biological explanation.
• Figure 4B shows that the BRCA and PRAD group is similar. This is very unlikely and the authors should provide much more evidence that Breast cancer and prostate cancer has very similar molecular features

Minor comments.

• Line 28: In the abstract transcriptomics is indicated as important factor to the integrated clustering, but this is mRNA and not miRNA. Please change it.
• Line 82: Could the authors be more specific what they used as input for the CNV analysis? Is this the Methylation data, SNP-array data or the DNA-seq data? Please clarify and show more details of the analysis.
• Line 88: The RNA data is normalized by the authors.
The method of normalization is not described and it is not clear why the authors did not use the normalized RNA data used and shared by CCLE (indicated with RSEM values).
• Please add the versions of all software packages that were used.
• Four single omics data clustered were built with 1-Pearson’s correlation coefficient distance metrics. Euclidean distance would reflect the amplitude of the data as well. Why did the authors not used the Euclidean distance as distance metrics for the clustering? Is this based on the output of NbClust? If so please provide the results.

Experimental design

see Basic reporting

Validity of the findings

see basic reporting

Reviewer 3 ·

Basic reporting

Although heatmaps like those shown in figure 1 are very popular, they do not really convey much information to the reader. Moreover, in this particular instance, clusters –that are the focus of the message- are barely distinguishable from the small color-coded boxes on the top of each panel. Color codes are good when dealing up to 6 or 7 entities, but they become confusing with larger sets. Numbering or lettering should be used instead. Moreover, each panel has a double color code (for instance, in Fig1A , "mRNA group" and "TCGA disease"), but only the latter has a legend; for the mRNA clusters there is not a color-key legend.
Overall, this kind of figures is not really informative and I strongly recommend swapping Figure 1 with Supplemental figure S1 (or something similar to it), where the relationship between the clusters found with the different techniques and the original classification is more evident. Before this, it is however mandatory a clear explanation of what the numbers in Supplemental Fig 1 exactly represents, and to omit all the instances of "-0.00", turning them in blank cells.
While in general the language is adequate, unambiguous and clear, some key definitions are missing (or well hidden). For example, it is not clear the definition of "domination" of a cancer type in a cluster. Cluster C6 (from integrated clustering analysis) is considered to be dominated by a single cancer type (SARC), while C5 by two types (HNSC-ESCA). However, by judging from the proportion of cancer lines in the pie charts in Fig2B, the two clusters seem quite similar in term of heterogeneity. The criterion for "domination" should be better explained.
Minor points about basic reporting :
Page 11 line 143: when reporting about GO enrichment, expressions like: "…these cell lines were upregulated in reproductive system development and morphogenesis of embryonic epithelium.", although understandable in the context, are a rather odd, and should be rephrased.
Page 13 Line 209: it is not clear how the high level of CDH1 in C3 cluster or the levels of VAV1 and STAT5 represent "an example" of what is stated in the previous sentence about the proportion of cancer type in two-cancer type dominated clusters.
Page 13 Line 212 "dominated type" should be "dominant type".

Experimental design

The main issue of the manuscript is about the scope of the study. Thorough multiplatform analyses on different cancer cell types have been already performed on patients samples, and to "extend" this kind of analysis to cell lines would not add much information about the biology of cancer by itself. Especially since many doubts about the reliability of cell lines have been casted repeatedly in the last decade.
The present study, however, could represent a valuable analysis for the reliability of available cell-lines as model of cancer. Rather than focusing on conclusion that have been already drawn by others researchers on more biologically relevant data, authors should focus more extensively on the comparison between the results of their analysis on cell lines and those obtained from patients. It would be useful, for example, to determine at which extent the cell lines resemble their (putatively) respective counterparts. In parallel, it would be of great interest to identify some "pan-cell-culture" artifacts, due to exposure to FCS or other tissue-culture media. This would be of great help in interpreting results from many studies performed on cell lines.

Validity of the findings

In general, the methods employed for the analysis are sound. It is rather surprising that all SKCM cell lines are strictly clustered together by many different kind of analysis, when melanoma is known to harbor a large variability. This aspect should be carefully discussed.

Additional comments

The manuscript from Yang and colleagues describes a meta-analysis of data derived from multiple high throughput techniques, on several cancer cell lines. Although the analysis has several interesting aspects, the manuscript should be modified to be suitable for publication. As detailed above, the major issues are two: the overall scope of the study, that should be refocused and the selection of the kind of figures to show in order to maximize the information conveyed.

---

## Round 0.2 · Major Revisions

The changes made to the original manuscript have not entirely addressed all the issues raised by the referees. Therefore, further improvemente is needed.

Reviewer 2 ·

Basic reporting

The manuscript has been revised thoroughly and improved significantly. Moreover, many reviewer questions have been properly addressed in the newest version of the manuscript. However, I have still some concerns regarding the current manuscript.

Major comments:

1. In the revised manuscript the figure legend are still very limited. For instance Figure 1 is not interpretable if you only read the figure legend.
2. In am still not convinced that the three PRAD cell lines ought to cluster with the BRCA cell lines. Although in both cancers drivers of genes in the Homologous repair system have been identified, such as BRCA2, their physiology is very different. In my opinion the number of PRAD cell lines is too limited and I expect when more PRAD cell lines could have been incorporated a distinct PRAD cluster would popup. Therefore, the conclusions in the manuscript based on the clustering of multiple tissues in a single cluster may be premature and based on underpowered data. Now, it is still hard to assess how reliable the clusters are.

Minor comments

1. line 145 “dada” should be changed to “data”

Experimental design

-

Validity of the findings

-

Additional comments

-

Reviewer 3 ·

Basic reporting

No comment (see below)

Experimental design

No comment (see below)

Validity of the findings

No comment (see below)

Additional comments

In my first report, the rigid form of PeerJ forced me to start my comments by writing about basic reporting, experimental design and validity of the findings, while -in this case- my main concerns were about the overall structure and focus of the manuscript (that I inserted in general comments). My impression is that this "unnatural" structure of the report prompted the authors to almost overlook what was my main concern: namely the scope of the study. While I acknowledge they modified their manuscript improving the definition of key concepts and improving the quality and the information conveyed by the pictures, however, the structure of the manuscript remains substantially unchanged, as clearly underlined by the abstract and the introduction. Therefore, I can't omit to repeat my previous comment:

"The main issue of the manuscript is about the scope of the study. Thorough multiplatform analyses on different cancer cell types have been already performed on patients samples, and to "extend" this kind of analysis to cell lines would not add much information about the biology of cancer by itself. Especially since many doubts about the reliability of cell lines have been casted repeatedly in the last decade.
The present study, however, could represent a valuable analysis for the reliability of available cell-lines as model of cancer. Rather than focusing on conclusion that have been already drawn by others researchers on more biologically relevant data, authors should focus more extensively on the comparison between the results of their analysis on cell lines and those obtained from patients. It would be useful, for example, to determine at which extent the cell lines resemble their (putatively) respective counterparts."

---

## Round 0.3 · accepted · Accept

The authors have adequately addressed the issues raised by the referees. Tha manuscript is now suitable for publication.

Reviewer 2 ·

Basic reporting

- As my major comments on the power of the analysis and the unclear figures are solved I have no additional comments or remarks

Experimental design

NA

Validity of the findings

NA